# Low-rank *Gallus gallus domesticus* chicks are better at transitive inference reasoning

Jonathan Niall Daisley [1], Giorgio Vallortigara [2] & Lucia Regolin [1✉]

A form of deductive reasoning, transitive inference, is thought to allow animals to infer relationships between members of a social group without having to remember all the interactions that occur. Such an ability means that animals can avoid direct confrontations which could be costly. Here we show that chicks perform a transitive inference task differently according to sex and rank. In female chicks, low-ranking birds performed better than did the highest ranked. Male chicks, however, showed an inverted U-shape of ability across rank, with the middle ranked chicks best able to perform the task. These results are explained according to the roles the sexes take within the group. This research directly links the abilities of transitive inference learning and social hierarchy formation and prompts further investigation into the role of both sex and rank within the dynamics of group living.

[1] Department of General Psychology, University of Padova, Padova, Italy. [2] Centre for Mind-Brain Sciences, University of Trento, Rovereto, Italy. ✉email: lucia.regolin@unipd.it

Animals that live in social groups will employ social cognition and social learning to interact successfully with conspecifics[1] in order to find food and shelter, avoid predators and gain access to mating and nesting opportunities. By extension, individuals will often need to recognise other individuals and to track their social patterns (dominance relationships etc.) as well as to determine their own status within the group[2,3]. In order to do this they will have to be capable of making inferences about their own status on the basis of observed interactions of other individuals and by doing so avoid direct confrontations which may lead to injury and potentially, death.

Chickens (*Gallus gallus domesticus*) live in social groups, usually consisting of a male bird with a number of females present in its territory; a small number of other males may also be present in the periphery of the territory. They form a generally linear social hierarchy known as a 'pecking order' in which the male bird is dominant over the females, whilst the social hierarchy shown by the females is determined by which birds dominate at pecking others and where they perch during roosting[4,5]. The hierarchy is maintained by continued interactions, usually displays of dominance, between the females. The ability to recognise other individual conspecifics is required for the formation and maintenance of the pecking order[6,7] and from this the hens can make inferences about their own status on the basis of observed interactions of other individuals. Indeed, it has been shown that chickens will avoid fighting a stranger that has been observed as being dominant over another individual known to be higher in the hierarchy[8].

Transitive inference is a form of deductive reasoning which has been demonstrated across the animal kingdom, including in mammals (e.g. rats *Rattus norvegicus*[9] and chimpanzees *Pan troglodytes*[10]), birds (in Pinyon jays *Gymnorhinus cyanocephalus*[11]; greylag geese *Anser anser*[12] and in domestic chickens[13]) and in fish (in the brook trout *Salvelinus fontinalis*[14]). It is thought that transitive inference allows animals to infer social relationships between individuals and may underpin the social cognition of hierarchy formation (e.g. a group's dominance hierarchy such that if A > B and B > C, then A > C). Transitive inference has been shown to occur readily in young domestic chicks[13], a precocial species that is known to form social hierarchies very early in life: aggressive interactions that relate social hierarchies may be apparent within the first week after hatching[15]. Performance in a transitive inference task was linked to brain lateralisation in the chick, such that those chicks that were forced to use their left hemisphere only to recall information by occlusion of the left eye were unable to perform the task, whilst those individuals able to use their right hemisphere (left eye in use and right eye occluded) performed the task[13]. This can be linked to the importance of the right hemisphere in a role in social cognition, including recognition of conspecifics[7,16]. In addition, females performed the task better than did males consistent with females forming stronger attachments than do males and potentially related to the fact that female chickens form stronger dominance hierarchies in groups of larger sizes[17].

Does social status influence an individual's ability to determine its place in the hierarchy and, indeed, perform a transitive inference task? A link between social rank and an individual's ability to perform transitive inference has never been directly investigated. In the experiment described below, we confronted 5 day old chicks that had been raised in isolation with a small number of conspecifics and ranked them according to the number of pecks they received and emitted to produce a dominance hierarchy. Subsequently, chicks were maintained in groups with the same individuals. Chicks were then trained to discriminate pairs of visual stimuli through associative learning, so that they could build a hierarchy of the stimuli. The chicks were tested on the same pairs presented during training and, in addition, they were presented with a pair of stimuli never previously experienced together but each of which had been rewarded to an equal level during training (i.e. BD, where B should be chosen as a demonstration of TI learning). The pair AE (A always rewarded during training, E never rewarded) was used as a control, in which birds should choose A. Results are discussed with regards to differences in performance that were apparent between the sexes and between the ranks within the sexes and how they relate to social mechanisms in chickens and to the relevance of transitive inference as a tool within social hierarchies. Namely, we show that in young chicks, lower-ranking individuals tend to outperform higher-ranking individuals in the transitive inference task and that the sexes appear to perform the task with different abilities according to rank, with lower-ranking females performing better than higher-ranked ones and middle-ranked males performing the task better than higher-ranked individuals.

## Results

**Training session analyses.** Following a one-sample Kolmogorov-Smirnov Test, training data (all pairs: AB, BC, CD and DE) were natural-log transformed (ln) to produce data with a normal distribution.

The number of trials required by the chicks to reach criterion (20 consecutively correct responses) was analysed by repeated measures ANOVA. The within-subject factor was the ln-transformed premise pair (AB, BC, CD and DE), the between subject factors were, sex and rank. Across the training trials (AB, BC, CD, DE) there was no significant difference between ranks with regards to number of trials to criterion (repeated measures ANOVA, $F_{6,114} = 0.810$, $p = 0.564$ $\eta^2 = 0.029$). A significant difference between the sexes was apparent, however (repeated measures ANOVA, $F_{3,114} = 8.677$, $p < 0.001$, $\eta^2 = 0.139$): females performing better than did the males, with fewer trials required across training for the pair CD (ANOVA, $F_{1,43} = 13,575$, $p = 0.001$, $\eta^2 = 0.193$). No differences were present for the other premise pairs AB (repeated measures ANOVA, $F_{1,43} = 0.897$, $p = 0.350$, $\eta^2 = 0.023$), BC (repeated measures ANOVA, $F_{1,43} = 1.331$, $p = 0.256$, $\eta^2 = 0.034$) or the pair DE (ANOVA, $F_{1,43} = 2.561$, $p = 0.118$, $\eta^2 = 0.029$). No difference was present in the interaction between rank and sex (repeated measures ANOVA, $F_{6,114} = 0.574$, $p = 0.750$, $\eta^2 = 0.029$). The total number of trials required to criterion across all training trials was not different between ranks (ANOVA, $F_{1,43} = 0.479$, $p = 0.623$, $\eta^2 = 0.025$) or between the sexes (ANOVA, $F_{1,43} = 1.144$, $p = 0.292$, $\eta^2 = 0.029$), however indicating that the training pair differences did not impact upon the overall outcome of training to criteria per se (further discussed within *Overall BD* section below).

**Rank analysis.** Following a 15 min exposure to conspecifics for the first time, chicks were classified as Rank 1 (highest number of pecks given to conspecifics, least received), Rank 3 (fewest number of pecks at conspecifics and most received) or Rank 2, those that showed an intermediate response. The total number of pecks at both conspecifics and to the environment (cage walls, water container) were analysed in order to determine as to whether pecking activity or directed pecking at conspecifics per se was the driver for the assignment of rank.

No differences were recorded in the number of pecks to the environment with regards to rank (ANOVA, $F_{2,43} = 2.641$, $p = 0.084$, $\eta^2 = 0.122$) or sex (ANOVA, $F_{1,43} = 1.589$, $p = 0.215$, $\eta^2 = 0.040$) with no interaction between these factors (ANOVA, $F_{2,43} = 1.214$, $p = 0.308$, $\eta^2 = 0.060$).

The total number of pecks to other chicks was significantly different between ranks (ANOVA, $F_{2,43} = 45.325$, $p < 0.0001$, $\eta^2 = 0.705$), between the sexes (ANOVA, $F_{1,43} = 29.77$, $p < 0.001$, $\eta^2 = 0.439$) with an interaction of rank and sex (ANOVA, $F_{1,43} = 6.029$, $p = 0.005$, $\eta^2 = 0.241$). Males produced more pecks to other male chicks than did female chicks to other females (males mean $= 18 \pm 3.3$; females $8.83 \pm 1.89$). Males assigned to rank 1 (highest ranked chicks) emitted $31.38 \pm 2.8$ pecks to other chicks during the assessment, rank 2 male chicks at a rate of $16.45 \pm 2.6$ pecks and the rank 3 (lowest rank) male chicks at a rate of $5.14 \pm 0.7$ pecks to others. On the other hand, rank 1 female chicks pecked other chicks with a mean score of $15.33 \pm 0.67$, rank 2 females $8 \pm 0.72$ and rank 3 females $3.17 \pm 0.45$. This directed pecking implies a stimulus-specific response rather than just a causation of rank being increased pecking, and possibly other behavioural, activity. The following were assigned Rank 1: 8 males; 6 females; Rank 2: 10 males and 6 females; Rank 3: 8 males and 6 females.

**Test session analyses**. Following a one-sample Kolmogorov-Smirnov Test, test data (stimulus pairs AE and BD) were natural-log transformed (ln) to produce data with a normal distribution.

Total AE response at test and the overall BD response were analysed to determine if the chicks were able to perform the tasks above chance, with rank and sex used as between subject factors.

**Training premise pairs during test**. Premise pairs AB, BC, CD and DE were presented across the testing sessions, as outlined in the methodology. Chicks performed above chance when presented with these pairings, at levels demonstrating strong memory for the premise pairs (AB mean $18.41 \pm 0.231$; BC $18.43 \pm 0.201$; CD $18.00 \pm 0.213$ and DE $17.95 \pm 0.181$; $t$-test, all $p < 0.0001$) with no significant difference recorded between ranks (ANOVA, $F_{2,43} = 0.143$, $p = 0.997$, $\eta^2 = 0.008$) or between sexes (ANOVA, $F_{1,43} = 0.297$, $p = 0.878$, $\eta^2 = 0.010$) and no interaction between rank and sex (ANOVA, $F_{2,43} = 1.255$, $p = 0.280$, $\eta^2 = 0.094$).

**Overall AE**. There was no significant difference between ranks in their response to AE (ANOVA, $F_{2,43} = 0.097$, $p = 0.908$, $\eta^2 = 0.005$). There was an effect of sex (ANOVA, $F_{1,43} = 4.787$, $p = 0.035$, $\eta^2 = 0.110$): males performed the task better than did females (mean $\pm$ s.d. males $19.77 \pm 0.43$; females $19.39 \pm 0.698$). There was no interaction of sex and rank (ANOVA, $F_{1,43} = 0.207$, $p = 0.814$, $\eta^2 = 0.009$), however.

**Overall BD**. A potentially confounding variable of the methodology employed during testing would be a bias in the number of positively rewarded B and D stimuli that chicks were presented with during the training sessions. During training the ratio of the number of rewarded presentations of B (B + in the premise pair BC) to non-rewarded presentations (B- in the presentation AB) was significantly different between the sexes (repeated measures ANOVA, $F_{1,43} = 4.938$, $p = 0.032$, $\eta^2 = 0.115$; male ratio 3.33 with sd 1.64, female ratio 2.43 with sd 0.67) but not between the ranks (repeated measures ANOVA, $F_{2,43} = 0.303$, $p = 0.740$, $\eta^2 = 0.016$) and with no interaction present between sex and rank (repeated measures ANOVA, $F_{2,43} = 0.107$, $p = 0.898$, $\eta^2 = 0.006$). Differences were also present between the number of rewarded D presentations (D + from premise pair DE) and non-rewarded D presentations (D- in premise pair CD) across training with a sex difference present (ANOVA, $F_{1,43} = 29.76$, $p < 0.001$, $\eta^2 = 0.439$; male ratio of 1.08 with sd 0.70 and females 2.21 with sd 0.66) but no differences between rank and with no interaction present (both $p > 0.100$). Analysis showed no difference in the number of

rewarded training stimuli (B + versus D+) between either the ranks (ANOVA, $F_{2,43} = 0.537$, $p = 0.589$, $\eta^2 = 0.027$) or between the sexes (ANOVA, $F_{1,43} = 0.602$, $p = 0.442$, $\eta^2 = 0.019$), however, with no difference in the total number of rewarded B presentations or total number of rewarded D presentations between the sexes (both $p > 0.05$). No correlation was present between final testing of BD (i.e between the number of 'correct' choices of B at test) and the ratio of rewarded B (B+) and rewarded D (D+) presentations during training (Pearson correlation ratio $\rho = -0.162$, $p = 0.293$, $n = 44$). Also, there was no correlation between total number of training trials to criterion and to test (BD) results (Pearson correlation ratio $\rho$ $-0.268$, $p = 0.079$, $n = 41$).

A significant difference was present at test (BD) between chicks of different rank (ANOVA, $F_{2,43} = 5.00$, $p = 0.012$, $\eta^2 = 0.22$) with regards to the number of correct choices recorded (refer Fig. 1). Post-hoc tests (Tukey HSD) pointed to differences between rank 1 and rank 2 chicks ($p = 0.012$) and between rank 1 and rank 3 chicks ($p = 0.044$).

There was a significant effect of sex at test (ANOVA, $F_{1,43} = 4.608$, $p = 0.038$, $\eta^2 = 0.08$) together with an interaction of sex and rank present (ANOVA, $F_{2,43} = 4.007$, $p = 0.026$, $\eta^2 = 0.21$) determining BD performance. Females showed a clear linear trend with individuals of lower ranks performing better than did individuals of higher rank at TI (overall between subjects effect of rank $F_{2,17} = 14.58$, $p < 0.001$; with differences present between females of all ranks: i.e. between rank 1 and rank 2 ($p = 0.023$), rank 1 and rank 3 ($p < 0.001$) and between rank 2 and rank 3 ($p = 0.012$)). There was no overall difference in variance between the sexes in BD performance, however (Levene's test, males mean$\pm$standard deviation, $14.04 \pm 2.2$; females $15.00 \pm 1.78$, $F_{1,25} = 2.36$, $p = 0.321$).

In males a significant difference was observed only between the highest ranking birds (which appeared to perform the test at a level not significantly above chance) and the middle ranking birds (rank 1v rank 2; $p = 0.034$, post-hoc) with the latter performing the task better.

**Discussion**
In the present study chicks were assigned a rank following a brief exposure to conspecifics of the same sex at 5 days post-hatch during which time pecking behaviours were recorded. These conspecifics had not been seen prior to this exposure. They subsequently lived in small groups with these same conspecifics (of 3 or 4 chicks) throughout the remainder of the experiment in which they were shaped and then trained to criterion (20 consecutive correct responses) on a series of paired-stimuli (different shapes of order A, B, C, D and E) across the following 15 days or so. Testing occurred following successful training, in which stimuli pairs were presented that had not been previously seen together. The test pair AE was of two stimuli one of which had always been rewarded throughout training (A) and the other never rewarded (E), successful discrimination of this pair showed a simple associative learning had taken place. Indeed, all chicks, of both sexes and all ranks, performed this task above chance.

The other test pair BD, was a stimulus-pair that had not been presented together previously and in which the two stimuli types had had a similar level of positive reinforcement to one another during training. Successful performance in this part of the test, by pecking stimulus B, represents the chicks' ability to have learnt the hierarchy A to E with all stimuli in between and the relationships between them suggesting that the chick has responded using a form of logical reasoning – transitive inference.

We have previously shown that chickens, when raised individually, are able to perform transitive inference[13]. In the present

**Choice of B in the comparison BD with respect to rank and to sex**

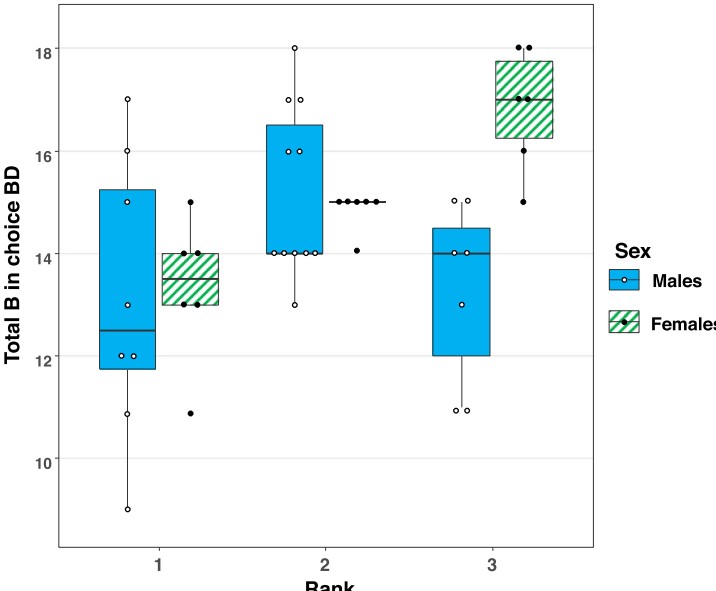

**Fig. 1 Choice of B in the comparison BD with respect to rank and to sex (total *n* = 44, 26 males and 18 females).** The box of the boxplot represent the 1st–3rd quartile of data with the 'whiskers' showing the maximum and minimum values. Values out with the maximum and minimum are outliers of the data set. Individual data points are shown as dots.

study, group-raised chicks were trained and tested in the equivalent manner and were shown to demonstrate learning. However, some caveats were introduced: the chick's position in the group's hierarchy and the sex of the chick were found to influence the learning outcome, namely the highest ranking male chicks performed worse at the task than middle-ranked birds and overall males performed less well than did females, with lower ranking females performing better than higher ranked females. We also confirmed that there was no direct effect of training per se on the testing outcome, with neither the total number of training trials completed, nor the ratio of reinforcements (between B+ and D+) determining BD performance at testing. The following expands on these findings and their potential implications.

Differences between the sexes were recorded during the training trials with the total number of rewarded presentations of B being higher in females compared to males with regards to the number of non-rewarded presentations of each premise pair and the number of rewarded D presentations greater in females than in males. There is no obvious reason for this difference, however, it can be speculated that, given that the pecking stimuli remained constant across all training for all individuals, something inherent in the stimuli (colour and/or shape) that the sexes respond to differently, may be present. Sex differences in colour preference (and familiarity with colours) has previously been described[18]. In addition, males and females show differences in shape preferences with males more attracted by unfamiliar patterns and females more attracted by familiar patterns[19]. It is likely that sex differences in brain lateralisation may play a role in the differences observed with right hemisphere/left hemisphere biases of different strengths present for the stimuli according to the sex of the chick. A left hemisphere visuospatial bias is present in males, but not females, for colour in a tidbitting paradigm and males tend to show hemisphere bias more prominently than females during observation of bead pecking in another individual[20]. It is possible, therefore, that training involving the placement of positively-rewarded stimuli to both the left and, subsequently, to the right of

the chicks is essentially more challenging for the male chicks given their inherent bias which may have been amplified by elements associated with the stimuli (colour and/or shape). Nevertheless, given that there was no difference in the number of rewarded training stimuli (B+ versus D+) between the sexes and that there was no correlation present between the number of 'correct' choices at test and the ratio of rewarded B (B+) and rewarded D (D+) presentations during the training phase as well as no correlation between total number of training trials to criterion and to test results, it is clear that the differences in training responses did not significantly influence the test results and conclusions drawn.

The recent ancestor of the chicken, the jungle fowl *Gallus gallus* lives in groups of several females and of one to a few males[21]. Within flocks, both male and female red junglefowl exhibit dominance hierarchies; dominant males tending to defend small mating territories from others while females within the flock will compete for food access. Dominance hierarchy formation is inherent in group living and indeed, living in groups appears to provide a predisposition for social facilitation and social learning behaviours. It is important for chicks to recognise their conspecifics from an early age and to be able to interpret the social interactions between them[22]. Young chicks engage in social learning during foraging, dust bathing and preening[23] and also during foraging, when social learning aids the chicks' ability to avoid noxious food items[24]. This becomes more apparent when agonistic activity begins and social hierarchies are formed - social learning ability per se is a requirement for the formation of dominance hierarchies. This ability is necessary for the chick and may underlie the fact that in chickens, dominance relationships may commence within the first week after hatching[15].

The ability to infer judgments on other individuals' ranks by observation only, and from this, the ability to infer its own rank status, means that an animal is able to predict the outcome of competition for resources (food, mating opportunities etc.) and thus avoid unnecessary and potentially injurious fights with other conspecifics[25]. In order to perform transitive inference

successfully this learning should be coupled with the ability to recognise others within the group, recognise their own interactions within the group context and assess the interactions between others within the group (although not all interactions are required to be memorised to determine the group social hierarchy).

Chickens have been shown to have excellent social cognitive abilities[26] with the ability of an individual to visually discriminate and recognise others in their social group[27] and to interact with them successfully by controlling its own actions and controlling processes that involve the interaction between itself and other members of the social group.

In the present study, differences in the ability to perform a transitive inference task were identified between the sexes, with females performing better than males overall. It is likely that such a difference is present due to the expression of social behaviours in group living that varies between male and female chickens. Many findings point to female chickens performing behaviours associated with group living and individual recognition better than do males. For example, female chicks are quicker and better able to discriminate familiar compared to unfamiliar conspecifics[20]. Female chicks' ability to perform transitive inference can be linked and added to these studies. We have previously shown a difference between the sexes in the ability to perform transitive inference[13]. Improved performance was demonstrated in females compared to males that had also been raised with an imprinting object only. This is consistent with female chicks forming stronger social bonds than do the males[28]. Also, females tend to live in more strict hierarchies and in groups of greater size[17]. Male chicks tend to interact more aggressively with conspecifics of the same sex, e.g. by eliciting more pecks at their social partners, than do female chicks and they also have a tendency to approach and stay with unfamiliar chicks more than do females[20].

Further to this, the performance of the chicks in the transitive inference task was linked to the rank of the chick as assigned during the initial presentation of its conspecifics. In male chicks, the highest ranked birds did not perform transitive inference significantly above chance level whereas the other ranks did. This could be because dominant males essentially do not need to learn a hierarchy given the relative simplicity of the dominance hierarchies formed in the species, rather they require only to 'dominate' all other males and protect their mating territories (this assumes that those chicks identified as being of the highest rank at day 5 of age proceed to remain so within the group across time). Conversely, the middle ranking males that did perform transitive inference, and are therefore likely to be able to assess other males within a hierarchy, may be of the level of rank whereby they will be able to challenge dominant males for territory and matings, or each other for e.g. satellite mating opportunities. The lowest ranked birds did perform above chance but appeared less able than the middle-ranked birds (although not statistically so). As speculated by Croney et al.[29] we suggest that 'submissive birds have to learn more relationships than dominant birds in order to avoid attacks by individuals to whom they are submissive. Likewise, they may have to be more mentally flexible so that they can adopt alternative strategies to obtain resources, such as food or mating opportunities', although their data did show a tendency for dominant birds, in a dyad, to perform better in a visual discrimination task.

This is in general agreement with Oden et al.[30] who showed that only the subordinate males form a dominance order, often due to being forced into a restricted space by more dominant, territorial birds. That the lowest ranking males in the present experiment appeared less adept at learning the task is in contrast to this, however. Whether this was due to there being no requirement to do so i.e. a case of knowing one's position to be the lowest and therefore no need to invest time and energy into learning the differences between the other ranks, or whether this was a case of chronic-type stress given that they were unable to escape the confines of their social environment, we cannot explicitly say (it is worth noting here that no demonstrable signs of feather-pecking e.g. feather loss from the head etc. were apparent within any of the groups).

In the present work, female chicks appear to demonstrate transitive inference ability linked in a linear manner to rank as assigned at day 5 post-hatch. As per male birds, the highest ranked appear not to invest in this form of learning in comparison to the other groups, although the highest ranking females do perform the task above chance. The lowest ranked female chicks, did however, perform significantly better than did the other ranked birds, potentially related to a requirement to continually monitor the status of other flock members either due to a greater potential to raise within the rank system themselves and/or due to a comparatively reduced 'stress' level in comparison to the male birds, the latter linked to females essentially living in groups as part of their species' social habits whilst males tend to be relatively solitary. It would therefore be most important for the lower ranked females to have an increased capacity to assess hierarchy and perform transitive inference, also potentially indicating a more stable dominance hierarchy in females in comparison to male birds, too. In addition, it has been shown previously that social learning may be linked to dominance rank in chickens: with hens learning more readily from a dominant individual as a demonstrator[31] whilst an individual hen's ability to learn may be related to its social rank (in Queiroz and Cromberg[32]). Indeed, work by Katz and Toll[33] found a correlation between dominance rank and the performance of chickens required to peck at every third piece of grain only, when presented to them. Further to this, it has been suggested that hierarchy formation may be influenced by group size, with larger groups showing reduced aggression towards one another and suggesting that chickens may adopt different social strategies according to group size, potentially meaning that being raised in larger groups may impinge upon transitive inference learning ability[34]. We would therefore agree with Marino[35] in that for 'chickens, as in other animals, social factors mediate learning factors in a complex way'.

It can only be inferred that the ranks assigned to the chicks at day 5 remained through to the testing stage. With an initial group of chicks we did carry out a repeat of the rank test but at day 18, prior to testing (not reported in the results section). A larger arena was used, due to the increase in size of the chicks, but almost no interactions were recorded in the time period (up to 30 min) used, suggesting that this test was not fit for purpose at this stage of chick development (or within the context of the social groupings that had developed across the intervening 13 days), at least for chicks that had habituated to one another. The use of a novel stimulus or food may have presented a behavioural challenge to elicit intra-group pecking but time constraints meant that this was not carried out. It should be noted that at 5-days post-hatch true aggressive behaviours have not been reported in chickens, although treatment with testosterone has been shown to elicit aggression in chicks as young as 3 days post-hatch[36]. In Burmese Red Jungle Fowl (*Gallus gallus spadiceus*) Kruijt observed the first aggressive peckings at 10 days of age, with juvenile fights not starting until three weeks of age[37]. However, the assessment of competition for a restricted food source has shown that social hierarchies exist within the first week of life[15]. In the present study, only water was provided in the local environment during the rank assessment phase and no direct competition for the resource was noted during the 15 min interaction period. The fact that those initial pecking behaviours appeared to have a persistent relationship to transitive inference

performance does suggest that they are related to rank. Ideally, either an initial or subsequent group test of the chicks at c. 10 days of age, with limited food resource, would have been preferable in the present work, however following previous work on transitive inference from this laboratory[13] with chicks being shaped from day 9 of hatch, it was considered potentially too disruptive to group chicks at the start of shaping to produce consistent results across training.

A further assumption, was that the chicks in the present experiment remained in the static hierarchy of individuals throughout the experiment as defined by their interactions on day 5 post-hatch (as referenced by the results obtained, we do suggest that this is indeed the case). However, Chase and Lindquist[38] have challenged the use of the type of assessment we have carried out (and assumed) in this paper, with the idea that such dominance is likely to be dynamic and could at some stages be non-linear. Our experimental set-up is based on the initial formation of rank with pair-wise, winner-loser encounters. We accept that given the chicks' age at this initial encounter and our premise that the interactions recorded at this initial phase were used to compute a static model, the implied ranking used here is not free from criticism. However, we would again point to the relatively consistent results, that are underlined by a rationale based on the relationship of the roles of the sexes, pointing to assigned rank being a driver of transitive inference performance in chicks. In addition, other experimenters[4], have derived transitive, and generally, linear dominance hierarchies in chickens.

It is suggested that there are implications for memory capacity in the ability to perform inference tasks and whether an immediate inference strategy or a transitive inference learning should occur[39] and that these forms of learning are dependent on costs and benefits for the winner and loser. The form of transitive inference learning in chicks in the present experiment is potentially of a reduced cost to the loser compared to the social interactions (including aggression) to which the chick would be subject to when presented with the form of learning in a group of conspecifics, given that the only consequence of an incorrect peck is that the chick is not rewarded with food. As such, it could be argued that the immediate inference strategy could be at play here. This would suggest, however, that chickens are performing the task using a different skill set (and memory capacity[39]) that is a correlate of that used when in a social setting; the results in the experiment can be explained in a way that relates to the ecology of the rank and sex of the individual performing the task (as outlined earlier). Further work, involving not just a food reward as here but rather a social reward (e.g. access to a preferred conspecific of a dyad) could disentangle this possibility.

In conclusion, behaviours that could be interpreted as being aggressive (pecks to the head and body of others) in chickens at a very young age are associated with an ability to perform a transitive inference reasoning task as the chicks get older. This ability is not a direct correlation with aggression, rather it is related to the sex of the chick: with those female chicks that show a reduced aggression during initial interactions performing transitive inference to a greater degree. We believe that transitive inference performance is therefore related to rank and sex of the chick and can be explained, at least to a greater degree, by the requirements of group living in which the sexes perform different roles. The performance in transitive inference learning is likely influenced by social learning factors

## Methods

**Subjects and rearing conditions**. Subjects were 44 chicks (*Gallus gallus domesticus*; 26 males and 18 females) from a Hybro stock (a local variety derived from the White Leghorn breed). All chicks hatched at the laboratories of Comparative Cognition at the University of Padova, from eggs obtained weekly from a commercial hatchery (Agricola Berica, Montegalda, VI-Italy). Upon arrival at the

laboratory, eggs were placed inside an incubator at a temperature of 37.7 °C and humidity of around 50–60%.

After hatching, chicks were assigned sex and were taken to be reared individually with food and water presented ad libitum. The cages were kept constantly (24 h/day) lit by fluorescent lamps. Temperature (29–30 °C) and humidity (68%) were controlled. The chicks were gradually food-deprived to between 80% and 90% of their *ad libitum* feeding weight; water was always available.

**Apparatus**. All parts of the experiment took place in a separate room (experimental room) located close to the rearing room.

The experimental apparatus consisted of a rectangular white-painted cage (60 cm long × 40 cm high × 33 cm wide), made of four uniformly white-painted wood panels, with an opening at the bottom of one of the smaller length walls to enable two food-boxes to be placed. Each food-box consisted of a green rectangular plastic box (12 cm long × 6 cm high × 5.5 cm wide) with a green drawer (13 cm long × 3 cm high × 5 cm wide) that could be pushed open by the experimenter in order to allow chicks to reach food contained within; on the top of the food-box there was a plastic envelope to permit insertion of the stimuli (at an angle of 45°). Above the apparatus an electric light bulb lit the environment.

**Stimuli**. All stimuli used in this experiment were printed on a rectangular piece of white paper (9 cm long × 5.5 cm high) and placed in a rectangular plastic display (9.5 cm long × 6 cm high) fixed on the bearing of the food-box.

**Assessment of rank**. On the 5th day post-hatch, chicks were taken from their holding apparatus to a cage (28 cm long × 32 cm high; illuminated with a 25 W bulb and with water ad libitum) where they were placed together in groups of 3 or 4 chicks of the same sex; these chicks not having been exposed to each other prior to this phase. A camera (SONY HDR-HC9E) was placed above the cage and the interactions of the chicks during 15 min were recorded. Chicks' pecking responses to each other and to the environment were collated. Simply, and in all cases, an individual was recorded to produce significantly more pecks to other individuals than it received (this was determined to be a Rank 1 individual) whilst in all cases an individual emerged which received significantly more pecks than it produced (the Rank 3 individual). Those chicks that received and emitted an intermediary number of pecks were ascribed as Rank 2 (for chicks in a group of 3, this was therefore a single chick, for chicks tested and subsequently maintained in a group of 4, two chicks were assigned Rank 2 status).

Following the exposure to each other, these chicks were maintained in groups in cages (35 cm long × 35 cm high × 45 cm wide) for the shaping and testing phases.

**Shaping session**. Shaping started when chicks were 9-days old (following Daisley et al.[13]); all chicks had been food-deprived for a few hours prior to starting a session. Chicks were trained to peck a black dot of diameter ∅ 4 mm, printed in the centre of the paper card, so as to obtain food from the experimenter-operated drawer. When the chicks pecked the stimulus, the drawer was opened to allow access to the food for a few seconds. This procedure was repeated until the chick was accustomed to the operation.

**Training session**. Training began from 12 days of age and ~24 h after the end of shaping; in this intervening period chicks were deprived of food, but water was always available. In the training sessions chicks were confronted with paired presentations of stimuli so as to learn a hierarchical order of the training-stimuli: A > B > C > D > E: Stimulus A was a pink circle; Stimulus B a brown rectangle; Stimulus C a light-blue rhombus; Stimulus D a green cross and Stimulus E a yellow triangle (refer Fig. 2).

During training two identical food-boxes were used separated by a white cardboard partition. The training stimuli were presented pair-wise (one stimulus on each food-box). When the stimulus was pecked the appropriate drawer was opened: one held a food reward (+), the other was empty (−). The chicks were trained on a series of trials involving multiple (four) and simultaneous discriminations. The stimulus pairs were presented in the order: A + B−, B + C−, C + D−, D + E−, ('+' means choice reinforced with food; and '−' not reinforced). Thus, the stimulus pair A vs. B was presented first to the chicks; if the chicks pecked at A they received food from the drawer of the food-box (opened by the experimenter), if chicks pecked B the drawer for this box was opened but this held no food reward.

Chicks progressed to the next step (B + C−) when they had pecked at the correct stimulus (A) for twenty consecutive trials (the stimuli were randomised as to their presentation to the left or the right-hand side of the apparatus). Chicks progressed to subsequent pairings by pecking at the correct stimulus for 20 consecutive trials until all stimulus pairs had been presented.

**Test session**. A test session started ~24–36 h from the end of the training sessions, when chicks were 18–20 days old. The stimuli were the same as those used for the training sessions. Stimuli were presented pair-wise using two identical food-boxes. The test was divided into four sessions, with the chick undergoing 2 sessions per day. Each test session consisted of twenty trials, of which ten were presentations of pairs of

## Premise Pair Stimuli

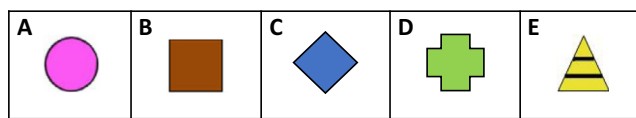

**Fig. 2 Training and testing stimuli. Stimuli were of different colours and shapes presented on cards of a uniform background.** Stimulus A a pink circle; Stimulus B a brown square; Stimulus C a blue diamond; Stimulus D a green cross and Stimulus E a yellow and black striped triangle. The hierarchical order of stimuli was maintained throughout the experiments such that A > B > C > D > E.

training-stimuli (i.e. AB, BC, CD and DE). In these trials chicks were food reinforced if they pecked at the correct stimulus of the presented pairs, otherwise no food was given if they pecked the incorrect stimulus (but, the food-less drawer was always opened). The other ten trials were of non-adjacent stimuli in the hierarchy: pair AE for five trials and pair BD for five trials. Choosing the correct stimulus (A) in the pair AE involved the ability to discriminate between a stimulus always reinforced (A) and a stimulus (E) that was never reinforced during training and the other trials of the test session, therefore AE represented a non-transitive pairing that had not been presented as a pair previously; responding correctly i.e. pecking at A should be considered as a demonstration of successful associative learning. The pair BD represented a test of transitivity, since in order to correctly discriminate the pair BD (i.e. pecking at B) chicks needed to have learnt the hierarchy, since B and D had been reinforced for the same number of trials as they had not been in the training pairs. The ability to perform this test correctly would suggest that the chick has responded using a form of representational learning, a form of learning likened to logical reasoning. In pairs AE and BD trials were given to extinction (neither drawer was opened, i.e. there was no consequence associated with the action of pecking the stimulus). Thus, the chicks were presented with a total of 80 trials over two days, including 40 trials of the presentation of pairs of stimuli that chicks had never been seen together: 20 trials of pair AE and 20 trials of pair BD. The position of the stimuli was balanced (left/right) across and within trials.

**Statistics and reproducibility**. Data analyses were carried out using the statistical package SPSS v.13, 2006 (released by SPSS Inc., 1968). Parametric statistics were used for the test results following a transformation (ln) to arrive at a normal distribution.

Statistical tests used throughout the manuscript were ANOVA (two-way), repeated measures ANOVA (e.g. where individuals were tested multiple times during the training regime) and Pearson corrleations (when determining the effect of training on the results at testing).

Effect sizes were determined using the $\eta^2$ - (Partial) Eta Squared method based on Cohen's F.

Levene's test was used to determine variances with regards to sex in the transitive inference response.

A total of 44 chicks were used in the experiments described: 26 males and 18 females. Chicks were grouped in 3's and 4's of the same sex such that a total of 8 male groups/replicates (8 rank 1 chicks, 10 rank 2 chicks and 8 rank 3 chicks) and 6 female groups (6 chicks of each rank 1–3) were formed. Training and testing of groups of chicks was carried out longitudinally, such that only a single group of chicks was being trained and subsequently tested prior to the start of the next group/replicate.

**Animal ethics**. All of the experiments were approved by the Italian and European Union directives on animal research, University of Padova License: CEASA (Comitato etico di Ateneo per la Sperimentazione Animale) prot. 37/2011, Ministry of Health License: 6/2012-B (10-01-2012). Animals used were chicks of both sexes (taken as eggs and hatched in situ) of the domestic chicken *Gallus gallus domesticus* of the "Hybro" strain. A total of 44 animals (26 males; 18 females) were used in the experiments described in this manuscript.

**Reporting summary**. Further information on research design is available in the Nature Research Reporting Summary linked to this article.

## Data availability

Data pertaining to Fig. 1 were uploaded to figshare with https://doi.org/10.6084/m9.figshare.13027829[40]. All other data are available from the corresponding author upon reasonable request. There are no restrictions on data availability.

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

## Acknowledgements

G.V. was supported from the European Research Council under the European Union's Seventh. Framework Programme (FP7) Grant ERC-2011-ADG_20110406, Project No: 461 295517, PREMESOR. G.V. and L.R. were supported by a PRIN 2017 ERC-SH4–A grant (2017PSRHPZ). This work was carried out within the scope of the project 'use-inspired basic research', for which the Department of General Psychology of the University of Padova has been recognised by the Italian Ministry of University and Research as a 'Department of Excellence' for the period 2018–2022. The authors wish to thank Federico Del Gallo and Tommaso Ioris for their help with animal testing and care.

## Author contributions

J.N.D., G.V. and L.R. designed research; J.N.D. performed research; J.N.D. analysed data; and J.N.D. G.V. and L.R. wrote and revised the paper.

## Competing interests

The authors declare no competing interests.
