## [Transparent Peer Review File · Communications Biology]

Reviewers' comments:

Reviewer #1 (Remarks to the Author):

The authors of the current paper have once again investigated transitive inference in chicks a few days old, built on their experiments and published paper of 2010 but with a different emphasis. Transitive inference (TI) is the process by which unknown relationships can be inferred from dyad characteristics (using $A > B$ and $B > C$ to infer $A > C$). In dominance hierarchies this TI ability is a more reliable and safer way of establishing a hierarchy than by trial and error. Hence, in animals that live in small or large groups and establish social hierarchies, this ability may be of great benefit and even aid survival. Indeed, the ability to infer dominance hierarchies by a deductive process is very widespread and has been demonstrated in many group living species be these primates, i.e. prosimian primates, squirrel monkeys or chimpanzees, in some birds, such as California scrub-jays, hooded crows, pinyon jays, and even in fishes such as brook trout, African cichlid fish and, more recently, also in the cleaner wrasse. In the current paper one misses some engagement with these latest findings.

1. Comments on missing information- expansions required in the Introduction and Discussion

The paper so far is barebone and one misses some engagement with the literature. Given that there are no particular limitations imposed by the journal, this is rather mystifying. One would hope for a good deal more information and explanation for a readership that may not be familiar with this topic. One would expect to find a wider engagement with the issues under investigation in the introduction and then again in the discussion in light of the results. For instance, hierarchy formation in group living species is an important topic. It is not made clear in the paper how hierarchy formation in this strain of chicks meshes with the experimental design. The experiments are started very early in the chicks' life, once the chicks are placed in the experimental framework. From memory, it would appear that chicks require some days before stable hierarchies are formed. I think the reader would want to know the species-specific details here as far as is known and how this experiment brings together hierarchy formation and TI.

2. Methods

It would also seem important to outline exactly what the rationale has been for the method employed and the days of shaping, training and testing so that one can be confident of the reasons for the choice of age at time of commencement of the tests. If developmental criteria are considered immaterial, that should perhaps also be spelled out. Chase and Lindquist (2016) have recently pointed out the fragility of explanations of social hierarchies in pecking orders. It would seem desirable to briefly allude to the literature and to specific points relevant to this paper.

3. Results

There is very little information and one misses at least one table of the actual recorded results of the chicks' performance during the experiments within the text. One gets no information on how individuals actually performed (some raw data have now been provided as S1). From Figure 1B, it would appear that the results for the males are more variable than those for the females. It would be worth testing this statistically.

4. Discussion

This section is particularly weak and the results for the females are not discussed, despite the fact that female show the clearest relationship between rank and transitive inference. Finally, Doi and Nakamaru (2018) pointed out that the immediate inference strategy (II), which estimates the opponent's strength based on the past history of the direct fights, is more likely to evolve in species with large memory capacity, while the TI strategy, which estimates the unknown

opponent's strength by transitive inference, can evolve and readily develop in species with a limited memory capacity.

The paper would benefit greatly from giving detailed explanations why the results are significant in the broader debate on a) the cognitive debate and b) as an essential adaptation. Further, two years ago, Brett Hayes and colleagues published a paper in the *J. of Experimental Psychology* called 'The Dimensionality of Reasoning: Inductive and deductive inference can be explained by a single process.' The authors were interested in uncovering whether the inductive or deductive process of arriving at a conclusion reflects qualitatively different cognitive processes. While their experiments have been conducted with humans, the argument of "Type 1" and "Type 2" reasoning are not new (Evans & Stanovich, 2013; Handley & Trippas, 2015; Kahneman, 2011). Although the precise characterization of these processes varies across different accounts, Type I is usually described as a "heuristic" form of processing that is fast, intuitive and proceeds with minimal demand on central processing resources. This is contrasted with Type II "analytic" processing that is slow, deliberative and requires working memory. Presumably, we are talking about Type I in this paper. It would probably go too far, on the basis of the results reported in this paper, to speculate as to whether the process can be described as rule-based (RB) or information-integration (II) learning or as described by Stephens and Kalish (2018) as perceptual category learning or item memory?

One would have hoped to see some more detailed comment on motivation for the lowest in the pecking order to have learned who 'A' was. I do not understand at all why the last lines of the conclusion state that it is 'counterintuitive' for the lowest in the hierarchy to show the best deductive reasoning—on the contrary. It would seem to me that such an ability is absolutely essential for the lowest in the queue to avoid complications or even death by misreading the hierarchy. Knowing how the system works would seem to be a distinct advantage, the less power an individual has.

5. Figures

Figure 1B: The label on the X-axis is confusing and not explained in the footnote.

6. Recommendation

Overall, the paper, as it currently stands, is simply not convincing to this reader because it seems incomplete both in depth as well as in discussion of the results especially for this journal. Neither the discussion nor the conclusions in any way offer a cogent argument why the findings are original and offer a substantial new finding with possible implications for our thinking about hierarchies or TI. The discussion does not even take in all the results. One feels that the paper does not deliver any punch lines or lead the reader into the fullness of the argument.

Perhaps one can encourage the authors to finish the paper and take extra care in this revision concerning all the points (1-5) raised so that this can become as exciting a paper as it can be because the results are indeed very interesting if not fully utilised so far.

Reviewer #2 (Remarks to the Author):

Behaviours of young chicks were examined in order to see if social rank (pecking order) is linked with the individual's cognitive capability of transitive inference. At post-hatch 5 days, individual rank was designated in terms of the (number of own pecks at others) minus (number of pecks received from others), and chicks were grouped into 3 ranks. Subsequently, from 9 to <18 days, chicks were trained for the transitive inference by using 5 visual stimuli (A to E), where order was set as A>B, B>C, C>D and D>E. Point is that B was not rewarded in the A>B training, but rewarding in the B>C, and so forth. On 18-20 days, the trained chicks were tested for the inference such as B>D, i.e., the pair that chicks have never experienced. As a control test, chicks were examined in the A>E test, where A was always rewarded but E not rewarded. The test results revealed a significant effect of social rank on the inference score, particularly in females. Males also showed a difference between high- and middle-

rank. The authors argue that the cognitive capability is ecologically attributable to the social dominance hierarchy.

This is a truly new finding on the possible ecological basis of cognitive development, particularly that the link was found in the young domestic chicks with limited post-natal experiences. I recommend the editor to accept this material for publication, after a few minor revisions and clarifications.

Minor comments

1. Is the social rank measured on 5 days unchanged until 18-20 days when the inference tests were done?
2. Is it possible that the so-called high rank chicks were simply highly impulsive in a sense of a lack of behavioural inhibition, which was associated with low learnability?
3. A concern about the equal numbers of training trials (as argued on p.7, line 164). The numbers of trials of B>C and C>D are expected to be equal in order to secure that B+/B- trials were equal to D+/D- trials. In this experiment (p.6, line 143-150), however, it seems that the chicks were initially trained by A>B, and if correct in 20 consecutive trials, they proceeded to B>C, and so forth. Numbers of these trials are therefore not equal, and so the ratio B+/B- trials could be simply higher than that of D+/D- in the lower rank chicks. Statistical examination must be added indicating that unequal numbers of trials do not explain the observed difference in the test scores.
4. Was the individual learnability (in terms of the number of training trials until the criteria of 20 consecutive correct response) not associated with the inference score?
5. P.6, line 139, "...were used separated by...", probably better delete "used".
6. In the test session (p.6, line 156-157), 10 trials were presentation of 4 types of pair (AB, BC, CD and DE). It is confusing because 10 divided by 4 is 2.5. Please clarify the numbers.
7. The y-axis of figure 1b says "Total BD", which should better be replaced with more appropriate term. In addition, chance level ("a random level, p. 3, line 69) must be indicated on this figure.

Reviewer #3 (Remarks to the Author):

The ms presents a single experiment that provides a direct test of social reasoning hypothesis positing that inferential abilities (i.e., transitive inference, or TI) developed as a result of social pressure. In that sense, the ms presents novel and potentially influential findings as the relationship between social rank and TI ability has not been explicitly explored previously. The authors report that lower ranking chicks were better at TI; they also reported that sex affected TI ability. Overall, I think this is a very promising line of research; however, the presentation of the data in the ms leaves many questions unaddressed. Although I understand that word restrictions make it difficult to include a detailed analysis, I am sure that at least some of the questions/concerns that I have could have been addressed in supplemental materials. Regardless of editorial decision, I strongly encourage the authors to pursue this line of research as it has a potential to address important questions in comparative research.

Major concerns:

- 1) The failure to show TI in the test could have occurred due to inability to learn the initial task or due to inability to infer that B is the correct choice in the pair BD. These possibilities need to be clearly distinguished and analyzed separately; we know that different species may have difficult time learning the task but once the task is learned, TI seem to inevitably follow (e.g., MacLean et al, 2008, Animal Behavior). The authors mention that lower-ranking males were unable to learn the task (line 82-83), but this statement is not supported by data or analyses.
- 2) Relatedly, it is not uncommon to find that animals can respond accurately in separately presented pairs (e.g., A+B- and B+ C-) but their performance falls apart once these pairs are presented together

(e.g., Lazareva & Wasserman, 2006, Behavioural Processes and many other pigeon reports). Thus, it is important to state whether chicks maintained high performance to training pairs during the test; if they did not, then the low TI could be attributed to incomplete learning rather than to any inferential failures.

3) Assuming the animals did learn original task to high levels of accuracy, is it possible that their BD performance has been affected by differences in associative values rather than by inference? See for example Lazareva et al., 2015, Hippocampus, where pigeons could rely on associative values or on inferential processes within the same training environment. One could make an argument that attention to differences in associative values could be dependent on the rank of individual.

Minor issues

4) I don't see any statement about normality of DVs; in my experience, proportion correct is rarely normally distributed and requires transformation prior to analyses. Alternatively, the authors could pursue a mixed-effect analysis with binomial link function that would avoid transforming the raw choices (0 and 1) into proportions.

5) How many males/females there were per each rank? Is it possible that some of the differences depicted on Figure 1b can be explained by low n's in some cells?

Reviewer #1 (Remarks to the Author):

The authors of the current paper have once again investigated transitive inference in chicks a few days old, built on their experiments and published paper of 2010 but with a different emphasis. Transitive inference (TI) is the process by which unknown relationships can be inferred from dyad characteristics (using $A > B$ and $B > C$ to infer $A > C$). In dominance hierarchies this TI ability is a more reliable and safer way of establishing a hierarchy than by trial and error. Hence, in animals that live in small or large groups and establish social hierarchies, this ability may be of great benefit and even aid survival. Indeed, the ability to infer dominance hierarchies by a deductive process is very widespread and has been demonstrated in many group living species be these primates, i.e. prosimian primates, squirrel monkeys or chimpanzees, in some birds, such as California scrub-jays, hooded crows, pinyon jays, and even in fishes such as brook trout, African cichlid fish and, more recently, also in the cleaner wrasse. In the current paper one misses some engagement with these latest findings.

1. Comments on missing information- expansions required in the Introduction and Discussion

The paper so far is barebone and one misses some engagement with the literature. Given that there are no particular limitations imposed by the journal, this is rather mystifying. One would hope for a good deal more information and explanation for a readership that may not be familiar with this topic. One would expect to find a wider engagement with the issues under investigation in the introduction and then again in the discussion in light of the results. For instance, hierarchy formation in group living species is an important topic. It is not made clear in the paper how hierarchy formation in this strain of chicks meshes with the experimental design. The experiments are started very early in the chicks' life, once the chicks are placed in the experimental framework. From memory, it would appear that chicks require some days before stable hierarchies are formed. I think the reader would want to know the species-specific details here as far as is known and how this experiment brings together hierarchy formation and TI.

We have expanded the introduction and largely the discussion to reference further works and tried to show a rationale as to why we did the experiment at the chicks' age that we did e.g. from line 296

2. Methods

It would also seem important to outline exactly what the rationale has been for the method employed and the days of shaping, training and testing so that one can be confident of the reasons for the choice of age at time of commencement of the tests. If developmental criteria are considered immaterial, that should perhaps also be spelled out.

Chase and Lindquist (2016) have recently pointed out the fragility of explanations of social hierarchies in pecking orders. It would seem desirable to briefly allude to the literature and to specific points relevant to this paper.

We have now outlined the rationale including why chicks were ranked at the age they were (refer line....). Also, the discussion now includes reference to Chase & Lindquist (lines from 296 and from 389)

3. Results

There is very little information and one misses at least one table of the actual recorded results of the chicks' performance during the experiments within the text. One gets no information on how individuals actually performed (some raw data have now been provided as S1).

The training and testing results have been expanded significantly to answer this comment. The supplementary data has been expanded to include results of pecking behaviours, training and testing results (AE, BD and premise pairs).

From Figure 1B, it would appear that the results for the males are more variable than those for the females. It would be worth testing this statistically.

A Levene's test of BD variance has now been carried out (the data showing a non-normal distribution). It appears that males and females do not show a difference in variance in BD performance, although, we agree, a tendency appears – line 159... .

4. Discussion

This section is particularly weak and the results for the females are not discussed, despite the fact that females show the clearest relationship between rank and transitive inference.

The discussion has been expanded and females discussed in contrast to male results.

Finally, Doi and Nakamaru (2018) pointed out that the immediate inference strategy (II), which estimates the opponent's strength based on the past history of the direct fights, is more likely to evolve in species with large memory capacity, while the TI strategy, which estimates the unknown opponent's strength by transitive inference, can evolve and readily develop in species with a limited memory capacity.

The paper would benefit greatly from giving detailed explanations why the results are significant in the broader debate on a) the cognitive debate and b) as an essential adaptation. Further, two years ago, Brett Hayes and colleagues published a paper in the J. of Experimental Psychology called 'The Dimensionality of Reasoning: Inductive and deductive inference can be explained by a single process.' The authors were interested in uncovering whether the inductive or deductive process of arriving at a conclusion reflects qualitatively different cognitive processes. While their experiments have been

conducted with humans, the argument of “Type 1” and “Type 2” reasoning are not new (Evans & Stanovich, 2013; Handley & Trippas, 2015; Kahneman, 2011). Although the precise characterization of these processes varies across different accounts, Type I is usually described as a “heuristic” form of processing that is fast, intuitive and proceeds with minimal demand on central processing resources. This is contrasted with Type II “analytic” processing that is slow, deliberative and requires working memory. Presumably, we are talking about Type I in this paper. It would probably go too far, on the basis of the results reported in this paper, to speculate as to whether the process can be described as rule-based (RB) or information-integration (II) learning or as described by Stephens and Kalish (2018) as perceptual category learning or item memory? One would have hoped to see some more detailed comment on motivation for the lowest in the pecking order to have learned who ‘A’ was. I do not understand at all why the last lines of the conclusion state that it is ‘counterintuitive’ for the lowest in the hierarchy to show the best deductive reasoning—on the contrary. It would seem to me that such an ability is absolutely essential for the lowest in the queue to avoid complications or even death by misreading the hierarchy. Knowing how the system works would seem to be a distinct advantage, the less power an individual has.

We have included reference to Doi and Nakamaru (2018) in the discussion; many thanks for the reviewer for pointing us in this direction. However, we would suggest that although the issues and points raised by the reviewer are most certainly of interest to the specialist reader we would not wish to be drawn into that level of speculation for what is, after all, a journal of biology rather than a journal of cognition.

The word and associated phrase that surrounds ‘counterintuitive’ has been withdrawn, as we agree with the reviewer that this may seem, in itself, counterintuitive to the reader!

5. Figures

Figure 1B: The label on the X-axis is confusing and not explained in the footnote.

The axis legend has been changed, as requested by the reviewers.

6. Recommendation

Overall, the paper, as it currently stands, is simply not convincing to this reader because it seems incomplete both in depth as well as in discussion of the results especially for this journal. Neither the discussion nor the conclusions in any way offer a cogent argument why the findings are original and offer a substantial new finding with possible implications for our thinking about hierarchies or TI. The discussion does not even take in all the results. One feels that the paper does not deliver any punch lines or lead the reader into the fullness of the argument.

Perhaps one can encourage the authors to finish the paper and take extra care in this

revision concerning all the points (1-5) raised so that this can become as exciting a paper as it can be because the results are indeed very interesting if not fully utilised so far.

We very much hope that the additions and amendments, as sought by the reviewers, will provide the appropriate answers required for potential publication.

Reviewer #2 (Remarks to the Author):

Behaviours of young chicks were examined in order to see if social rank (pecking order) is linked with the individual's cognitive capability of transitive inference. At post-hatch 5 days, individual rank was designated in terms of the (number of own pecks at others) minus (number of pecks received from others), and chicks were grouped into 3 ranks. Subsequently, from 9 to <18 days, chicks were trained for the transitive inference by using 5 visual stimuli (A to E), where order was set as A>B, B>C, C>D and D>E. Point is that B was not rewarded in the A>B training, but rewarding in the B>C, and so forth. On 18-20 days, the trained chicks were tested for the inference such as B>D, i.e., the pair that chicks have never experienced. As a control test, chicks were examined in the A>E test, where A was always rewarded but E not rewarded. The test results revealed a significant effect of social rank on the inference score, particularly in females. Males also showed a difference between high- and middle-rank. The authors argue that the cognitive capability is ecologically attributable to the social dominance hierarchy.

This is a truly new finding on the possible ecological basis of cognitive development, particularly that the link was found in the young domestic chicks with limited post-natal experiences. I recommend the editor to accept this material for publication, after a few minor revisions and clarifications.

Minor comments

1. Is the social rank measured on 5 days unchanged until 18-20 days when the inference tests were done?

Reference is now made of this in the discussion, whereby we did attempt to determine rank at the age of c. 18 days but were not successful. Further work is suggested that could add to our results and confirm (or not) a stable hierarchy.

2. Is it possible that the so-called high rank chicks were simply highly impulsive in a sense of a lack of behavioural inhibition, which was associated with low learnability?

Although number of pecks in total recorded during the rank assessment was higher overall in the higher ranked chicks, it was apparent that this was due to an increase in directed pecking towards conspecifics and not the local environment – i.e. there was a social explanation for this result and it was not due to behavioural inhibition (refer lines 107-118)

3. A concern about the equal numbers of training trials (as argued on p.7, line 164). The numbers of trials of B>C and C>D are expected to be equal in order to secure that B+/B- trials were equal to D+/D- trials. In this experiment (p.6, line 143-150), however, it seems that the chicks were initially trained by A>B, and if correct in 20 consecutive trials, they proceeded to B>C, and so forth. Numbers of these trials are therefore not equal, and so the ratio B+/B- trials could be simply higher than that of D+/D- in the lower rank chicks.

Statistical examination must be added indicating that unequal numbers of trials do not explain the observed difference in the test scores-

Scores were checked and no correlation was found between the number of reinforcements of B and D in relation to test BD results (as per lines 144 and 189 and on)

4. Was the individual learnability (in terms of the number of training trials until the criteria of 20 consecutive correct response) not associated with the inference score?

Statistics now presented show this not to be the case with no correlation between number of training trials and BD response (line 147)

5. P.6, line 139, "...were used separated by...", probably better delete "used".

Ok

6. In the test session (p.6, line 156-157), 10 trials were presentation of 4 types of pair (AB, BC, CD and DE). It is confusing because 10 divided by 4 is 2.5. Please clarify the numbers.

This has now been corrected to read 'true'

7. The y-axis of figure 1b says "Total BD", which should better be replaced with more appropriate term. In addition, chance level ("a random level, p. 3, line 69) must be indicated on this figure.

Figure legend amended. In addition, we have carried out further statistics to point to the fact that all ranks and sexes do perform above chance (as per Student's t-test). We would wish to thank the reviewer to bringing this to our attention and apologize for the rather slack, 'by-eye', look that we had previously carried out. We have not included a chance level line on the graph, given that we believe that such would obfuscate as the differences in sample sizes between ranks and sexes produce different potential variances that should be shown in addition to the 'chance line' in order to make sense of the results.

Reviewer #3 (Remarks to the Author):

The ms presents a single experiment that provides a direct test of social reasoning hypothesis positing that inferential abilities (i.e., transitive inference, or TI) developed as a result of social pressure. In that sense, the ms presents novel and potentially influential findings as the relationship between social rank and TI ability has not been explicitly explored previously. The authors report that lower ranking chicks were better at TI; they also reported that sex affected TI ability. Overall, I think this is a very promising line of research; however, the presentation of the data in the ms leaves many questions unaddressed. Although I understand that word restrictions make it difficult to include a detailed analysis, I am sure that at least some of the questions/concerns that I have could have been addressed in supplemental materials. Regardless of editorial decision, I strongly encourage the authors to pursue this line of research as it has a potential to address important questions in comparative research.

Major concerns:

1) The failure to show TI in the test could have occurred due to inability to learn the initial task or due to inability to infer that B is the correct choice in the pair BD. These possibilities need to be clearly distinguished and analyzed separately; we know that different species may have difficult time learning the task but once the task is learned, TI seem to inevitably follow (e.g., MacLean et al, 2008, Animal Behavior). The authors mention that lower-ranking males were unable to learn the task (line 82-83), but this statement is not supported by data or analyses.

This is largely covered in the statistics outlined in response to referee 2. We have changed reference to lower ranking males not able to learn the task

2) Relatedly, it is not uncommon to find that animals can respond accurately in separately presented pairs (e.g., A+B- and B+ C-) but their performance falls apart once these pairs are presented together (e.g., Lazareva & Wasserman, 2006, Behavioural Processes and many other pigeon reports). Thus, it is important to state whether chicks maintained high performance to training pairs during the test; if they did not, then the low TI could be attributed to incomplete learning rather than to any inferential failures-

The statistics show a performance that is consistent with training levels during the test phase of premise pairs – a direct comparison cannot be carried out as training was to criterion (20 correct responses in a row) whilst test was the presentation of 20 trials only.

3) Assuming the animals did learn original task to high levels of accuracy, is it possible that their BD performance has been affected by differences in associative values rather than by inference? See for example Lazareva et al., 2015, Hippocampus, where pigeons could rely on associative values or on inferential processes within the same training

environment. One could make an argument that attention to differences in associative values could be dependent on the rank of individual.

We have, hopefully, shown that given the no significant difference between B+ and D+ representations in training in relation to final BD test results, that there is no direct affect of associative value on test result (as per line 143 and on)

Minor issues

4) I don't see any statement about normality of DVs; in my experience, proportion correct is rarely normally distributed and requires transformation prior to analyses. Alternatively, the authors could pursue a mixed-effect analysis with binomial link function that would avoid transforming the raw choices (0 and 1) into proportions.

Data were transformed (by natural log – ln) – and this is now recorded within the text as such (e.g. line 122).

5) How many males/females there were per each rank? Is it possible that some of the differences depicted on Figure 1b can be explained by low n's in some cells?

Number of individuals in each rank now noted – (6 in each rank for females, 8-10 for males; line 116). We would have to agree that a larger sample size would have been preferred. However, both time and space constraints at the time of the experiments were in evidence. Nevertheless, we stand by the data and the results and conclusions that lead on from them and presented within this manuscript.

Reviewers' comments:

Reviewer #1 (Remarks to the Author):

Minor points

Lines 35/6—'also' be present in the periphery of the territory 'as well'.

('also' and 'as well' are the same thing-pls drop one or the other)

Lines 340 and 441 report the same information (number of chicks used)- best to cut in Line 441 and keep in Line 340 (under Subjects and rearing conditions where one would expect the information)

Reviewer #2 (Remarks to the Author):

I appreciate that the authors appropriately addressed to the issues that I raised. The revised MS is improved, and I recommend the editor to accept this for publication.

Reviewer #3 (Remarks to the Author):

Although I appreciated the authors' effort in adding more details and statistical analyses throughout, some of the descriptions were confusing and/or incomplete, and that made it difficult to evaluate the authors' conclusions. I am particularly concerned about the use of Fisher's LSD test throughout the ms, because this test only protects against type I error inflation only for 3 groups, and the authors have 6 groups (3 ranks x 2 sexes). The analyses should be redone with a more appropriate post-hoc test (e.g., Tukey's HSD) before any firm conclusions can be made.

Training analyses section

I disagree with the authors' interpretation here. Having a significant effect of sex in repeated measures ANOVA clearly suggests that both sexes are not performing in equivalent manner, despite the statement in the last sentence of this section. I'd also like to see a clearer attribution of statistics here; line 195 is, I suspect, a planned comparison of CD accuracy between males and females but I shouldn't be left guessing. I would also like to see the results of planned comparisons for the other 3 pairs. The analysis in lines 96-98 seem to be redundant; if I understand correctly, it is simply the analysis omitting within-subject effect of pair.

Rank analysis section

This section needs some explanation. I am assuming DV is the number of pecks (normally distributed?), but what are the IVs? A brief explanation of the logic behind looking at the pecks to chicks vs. environment would be helpful as well.

The analysis with total number of pecks seem to be redundant; results of analyses with pecks to the environments and pecks to chicks tell a complete story.

Overall AE section

The reported difference in means (males 19.77 vs females 19.39) seem to be awfully small for the reported F-value of 4.87. Are the values reported correctly here?

Overall BD section

Again, I'd like to see a brief rationale, and an explanation of what was used as an index of associative value. In any case, number of rewarded trials alone is not necessarily the best indicator of associative strength, because non-reward decreases the value of the stimulus and therefore modulates the final value. I would like to see a reward/nonreward ratio instead (B+/B- and D+/D-) or similar (see Weaver, Steirn, & Zentall, 1997, or any of Lazareva's transitive inference publications). Total number

of trials to criterion is not a good indicator here and can be dropped.

line 150: What post hoc tests? Please specify.

Supplemental data

I would like to commend authors on providing their full data set. A couple of suggestions: Please consider uploading .sav file as .xls or .txt for a wider availability. For the .xlsx file, consider adding a note explaining the meaning of different columns, some of which are not necessarily intuitive. For example, I understand the meaning of B+ column, but what is the meaning of B column?

Minor comments

line 87-88, and elsewhere: you are log-transforming accuracy, not pairs

line 126: I suggest changing the heading to _Training premise pairs during test_.

line 184: "transitive inference learning" is an odd term; while one needs to learn premises, the process of transitive inference itself is presumably different from the premise learning. I suggest using "transitive inference" term instead.

line 235-236: I would suggest clarifying that this is only true for dominant chicken as their hierarchy is relatively simple comparing to that of, say, primates.

We'd like to thank the reviewers, most especially Reviewer #3 for providing further feedback with regards to the submitted manuscript.

The following provides comment to the individual remarks from Reviewer #3 point by point.

Reviewer #3 (Remarks to the Author):

Although I appreciated the authors' effort in adding more details and statistical analyses throughout, some of the descriptions were confusing and/or incomplete, and that made it difficult to evaluate the authors' conclusions. I am particularly concerned about the use of Fisher's LSD test throughout the ms, because this test only protects against type I error inflation only for 3 groups, and the authors have 6 groups (3 ranks x 2 sexes). The analyses should be redone with a more appropriate post-hoc test (e.g., Tukey's HSD) before any firm conclusions can be made.

The reviewer is quite correct. Further analyses have been carried out using the appropriate Tukey's HSD as the post-hoc test.

Training analyses section

I disagree with the authors' interpretation here. Having a significant effect of sex in repeated measures ANOVA clearly suggests that both sexes are not performing in equivalent manner, despite the statement in the last sentence of this section. I'd also like to see a clearer attribution of statistics here; line 195 is, I suspect, a planned comparison of CD accuracy between males and females but I shouldn't be left guessing. I would also like to see the results of planned comparisons for the other 3 pairs. The analysis in lines 96-98 seem to be redundant; if I understand correctly, it is simply the analysis omitting within-subject effect of pair.

Yes, the statement was incorrect and has been changed to reflect the further analyses that were undertaken as per the reviewer's request.

Further analyses of the training premise pairs were carried out and did show sex differences (although no rank and rank x sex interactions were present, and those differences in training were not causally related to the 'correct' choice of B in the testing phase). These differences are discussed briefly within the manuscript.

Rank analysis section

This section needs some explanation. I am assuming DV is the number of pecks (normally distributed?), but what are the IVs? A brief explanation of the logic behind looking at the pecks to chicks vs. environment would be helpful as well.-

An explanation is provided within the text – essentially we wished to show that directed pecks to other chicks were not just a 'by-product' of overall activity levels per se.

The analysis with total number of pecks seem to be redundant; results of analyses with pecks to the environments and pecks to chicks tell a complete story.-

We agree with the reviewer and this has now been taken out of the ms.

Overall AE section

The reported difference in means (males 19.77 vs females 19.39) seem to be awfully small for the reported F-value of 4.87. Are the values reported correctly here?

This does appear to be correct following further investigation.

Overall BD section

Again, I'd like to see a brief rationale, and an explanation of what was used as an index of associative value. In any case, number of rewarded trials alone is not necessarily the best indicator of associative strength, because non-reward decreases the value of the stimulus and therefore modulates the final value. I would like to see a reward/nonreward ratio instead (B+/B- and D+/D-) or similar (see Weaver, Steirn, & Zentall, 1997, or any of Lazareva's transitive inference publications). Total number of trials to criterion is not a good indicator here and can be dropped.

The reward/non-reward ratio, as well as totals, were analysed. As noted above, differences were apparent between the sexes in the training phase. These did not correlate with the testing phase choice of correct response, however. The lack of any correlation does imply that, irrespective of the sex differences in ratio and total during the training phases, chicks of both sexes had 'arrived' at the testing phase with a relatively similar chance of being able to perform the task. This would point to the 'training to criterion' methodology used in the present study as being appropriate for both sexes, although the route/method that the chicks arrived at the criterion level may have been influenced by sex.

line 150: What post hoc tests? Please specify.

Supplemental data

I would like to commend authors on providing their full data set. A couple of suggestions: Please consider uploading .sav file as .xls or .txt for a wider availability. For the .xlsx file, consider adding a note explaining the meaning of different columns, some of which are not necessarily intuitive. For example, I understand the meaning of B+ column, but what is the meaning of B column?

This has been done.

Minor comments

line 87-88, and elsewhere: you are log-transforming accuracy, not pairs

line 126: I suggest changing the heading to _Training premise pairs during test_.

line 184: "transitive inference learning" is an odd term; while one needs to learn premises, the process of transitive inference itself is presumably different from the premise learning. I suggest using "transitive inference" term instead.

line 235-236: I would suggest clarifying that this is only true for dominant chicken as their hierarchy is relatively simple comparing to that of, say, primates.

All minor comments corrected as required.

Many thanks.

REVIEWERS' COMMENTS:

Reviewer #3 (Remarks to the Author):

All of my concerns have been adequately addressed in this revision. I congratulate the authors on a fine addition to the literature.